# Protective Effects of Filtrates and Extracts from Fungal Endophytes on *Phytophthora cinnamomi* in *Lupinus luteus*

**DOI:** 10.3390/plants11111455

**Published:** 2022-05-30

**Authors:** Carlos García-Latorre, Sara Rodrigo, Oscar Santamaria

**Affiliations:** 1Department of Agronomy and Forest Environment Engineering, University of Extremadura, Avda. Adolfo Suárez s/n, 06007 Badajoz, Spain; carloslatorre5@gmail.com (C.G.-L.); saramoro@unex.es (S.R.); 2Department of Plant Production and Forest Resources, University Institute for Research in Sustainable Forest Management (iuFOR), University of Valladolid, Avda. Madrid 57, 34004 Palencia, Spain

**Keywords:** endophytic fungi, bioprotection, plant growth promotion, metabolites, *dehesas*, *Phytophthora cinnamomi*

## Abstract

Fungal endophytes have been found to protect their hosts against multiple fungal pathogens. Frequently, the secondary metabolites produced by the endophyte are responsible for antifungal activity. To develop new bio-products that are more environmentally friendly than synthetic pesticides against *Phytophthora cinnamomi*, a serious pathogen of many plant species, the antifungal activity of filtrates or extracts from four endophytes was evaluated in different in vitro tests and in plants of *Lupinus luteus*. In the dual culture assays, the filtrate of one of the endophytes (*Drechslera biseptata*) completely inhibited the mycelial growth of the pathogen. Moreover, it showed a very low minimal inhibitory concentration (MIC). *Epicoccum nigrum*, an endophyte that also showed high inhibitory activity and a low MIC against *P. cinnamomi* in those two experiments, provided a clear growth promotion effect when the extracts were applied to *L. luteus* seedlings. The extract of *Fusarium avenaceum* also manifested such a promotion effect and was the most effective in reducing the disease severity caused by the pathogen in lupine plants (73% reduction). Results demonstrated the inhibitory activity of the filtrates or extracts of these endophytes against *P. cinnamomi*. A better insight into the mechanisms involved may be gained by isolating and identifying the metabolites conferring this inhibitory effect against this oomycete pathogen.

## 1. Introduction

*Dehesas* are the most important agrosilvopastoral systems in the southwest of the Iberian Peninsula, originated by the progressive clearing of the traditional Mediterranean forest of *Quercus* spp., maintaining around 20–60 trees per hectare [1,2]. This system is mainly focused on extensive livestock grazing, supplemented by the exploitation of other secondary resources such as hunting, firewood, charcoal, or cork. This multipurpose way of exploitation, together with the important ecosystemic services it provides, might explain the high environmental value of *dehesas*, which has allowed a more sustainable use of the scarce availability of natural resources [3].

However, due to the harshness of the edaphoclimatic conditions, the natural pastures of *dehesas* present limited herbage yield and high interannual variability [4], resources often insufficient to feed the livestock. For this reason, farmers are traditionally forced to grow different forage crops, such as oat (*Avena sativa*), occasionally in combination with vetch (*Vicia sativa*), or wheat (*Triticum aestivum*) to supplement the natural pasture production [5,6]. Within those supplementing crops, the use of grain legumes is increasingly growing [7] due to their capacity to improve soil structure, fixation of atmospheric nitrogen, and the general soil situation, resulting in an additional production increase in the mid- and long-term [8].

For this purpose, yellow lupine (*Lupinus luteus* L.), cv. “Tremosilla”, has traditionally been used in *dehesas* [9], due to its adaptation to the Mediterranean climate and to the ability to thrive in the acid soils of *dehesas* [10]. Furthermore, lupine species are among the legumes with the highest protein concentrations in their forage, especially yellow lupine, which has a protein content of about 40% and one of the lowest concentrations of toxic alkaloids (1.07 mg per 100 g) [8]. This makes them a very valuable source of nutrients for the livestock feed, with 30–40% non-starch polysaccharides and 5–15% oil [10,11]. In addition, yellow lupine seeds contain other very interesting compounds for feeding, such as polyphenols and bioactive peptides [12], and their content in cysteine and methionine is double the amount other lupine species have [13].

However, *L. luteus* has been shown to be susceptible to *Phytophthora cinnamomi* [6], an oomycete associated with root rot and holm oak decline [14,15,16], which has been commonly found to be a serious problem in *dehesas* for the tree layer [17]. In yellow lupine, this pathogen produces the wilting of the aerial part and root rot, which leads to plant death [18]. Therefore, in addition to the direct damage to the lupine plants that this pathogen could cause, they may be a potential source of inoculum as the amount of *Phytophthora* in soil could highly increase after the attack on lupines, which could lead to more serious damages to the already quite affected holm oak trees. These aspects could be important limitations for the extensive use of *L. luteus*. Thus, the development of methods that allow the protection of the crop and reduce the negative impact of this oomycete in lupines and consequently in the tree layer, are essential. The few effective active substances available against *P. cinnamomi* [19] and the current restrictions on the use of synthetic chemical treatments in agroforestry systems, limit the control of the pathogen. For these reasons, the development of biocontrol techniques against the pathogen based on the use of beneficial microorganisms could play a key role in integrated disease management. In this context, endophytic fungi can be an important source of biocontrol agents, to enhance the viability and sustainability of this agroforestry system.

Among the wide biodiversity found in the kingdom of Fungi, endophytes are species capable of growing asymptomatically and colonizing the internal plant tissues [20]. These species play an important role within plant ecosystems, associated with the symbiotic benefits that many of them produce in their host, especially under stress conditions [21,22,23]. Among such benefits, the protection of their host against a wide variety of harmful organisms is one of the most important actions performed by endophytes [24,25]. In the specific case of *P. cinnamomi*, there are several studies that have already shown the potential of different fungal species, such as *Trichoderma* sp., to be used as biocontrol agents against this pathogen in vitro [17,26].

When a symbiotic relationship between plant–endophyte is established, the endophyte obtains nutrients and protection and, in turn, it supports the development of its host, usually by synthesizing certain secondary metabolites that favor the survival of the plant [27]. These secondary metabolites represent an important source of active compounds [28,29] that can be stimulated by the complex interaction established with the host [30]. However, this interaction between endophytes and plants is essentially controlled by the genes of both individuals and modulated by the environment [31]. This is the reason why many of these positive effects found in the experimental assays are so inconsistent once the living endophyte is inoculated directly in-field conditions, as the response may change depending on the signals received. This might be a serious limitation for the massive use of fungal endophytes as biocontrol agents (BCA) with commercial significance.

To address this limitation, we propose the direct use of the metabolites responsible for the bio-protection, instead of the living organism, according to the hypothesis that such metabolites, previously produced under controlled conditions in vitro, may avoid or limit the effect of the environment and genotypes interaction to obtain the desired protection. As another important advantage, the handling, formulation, and application of a metabolite might be much easier and might not require any sophisticated equipment in comparison with that of a living organism. To achieve our objective, we first need to identify endophytic strains that provide biocontrol and are associated with compounds that may be effective in limiting the pathogen’s ability to invade plant tissues. Therefore, according to this hypothesis, the objective of the present study was to evaluate if the filtrates and extracts (containing the secondary metabolites) of four endophytic fungi, previously isolated from healthy plants from *dehesas* of Extremadura, present antifungal activity against *P. cinnamomi* and if their application in *L. luteus* seeds and seedlings protect them against the pathogen.

## 2. Results

### 2.1. In Vitro Experiments

#### 2.1.1. Effect of Filtrates on the Pathogen Mycelial Growth

The split-plot ANOVA analysis showed the endophyte filtrate (degrees of freedom, *df* = 4; *F*-value, *F* = 2204.35; *p*-value, *p* < 0.0001), the measurement time (*df* = 4; *F* = 2681.10; *p* < 0.0001) and their interaction (*df* = 16; *F* = 158.26; *p* < 0.0001) to be all significant variables (*p* < 0.05). In general terms, the filtrate of three endophytes, E063 (*Mucor hiemalis*), E064 (*Drechslera biseptata*), and E179 (*Epicoccum nigrum*), was able to limit the mycelial growth in vitro of *P. cinnamomi*. The most impressive result was obtained by the filtrate of *D. biseptata* (E064), which caused complete inhibition of the pathogen growth during the duration of the experiment (Figure 1). The filtrate of *E. nigrum* (E179) produced, on average, a reduction of around 22% of the mycelial growth of the pathogen in comparison with the non-treated control.

#### 2.1.2. Minimum Inhibitory Concentration (MIC) of Endophytic Filtrates and Extracts

The filtrate of two of the endophytes inhibited the germination and growth of the pathogen in the MIC assay. While in the case of endophyte E179 (*E. nigrum*), the use of the non-diluted filtrate was necessary to obtain such an inhibition, for the endophyte E064 (*D. biseptata*) only half the doses were needed to achieve this inhibition (Table 1). When the extract was tested instead of the raw filtrate, all the endophytes were able to inhibit the germination and growth of the pathogen after a 48-h period (Table 1), but the endophyte E063 (*M. hiemalis*) showed the highest activity since it needed half of the extract dose (150 μg mL^−1^) in comparison with the other endophytes (300 μg mL^−1^).

#### 2.1.3. Effect of Extracts on the Lupinus Seeds Germination

In presence of the pathogen *P.*
*cinnamomi*, the germination of *L. luteus* seeds was significantly affected by the endophytic extract treatment (*df* = 5; *F* = 236.68; *p* < 0.0001), the measurement day (*df* = 10; *F* = 128.04; *p* < 0.0001) and their interaction (*df* = 50; *F* = 4.10; *p* < 0.0001), according to the split-plot ANOVA. The germination percentage of *L. luteus* without neither pathogen nor extract treatment reached almost 60% after 15 days, a value which decreased up to 23% in presence of the pathogen (Figure 2). After 15 days of incubation, the extract of all the endophytes limited the negative impact of *P. cinnamomi* on seed germination. On the last day of measurement, the extract of the endophyte E179 (*E. nigrum*) completely reversed the negative effects of the pathogen as it produced germination percentages statistically similar to that obtained in the control without the pathogen. Nevertheless, the effect of the extracts of all of the endophytes on the seed germination was quite effective in all the cases, as no significant differences were observed among them.

### 2.2. In Planta Experiments

#### 2.2.1. Effect of the Pre-Emergence Application of Extracts on *Lupinus* Seedlings in Growth Chamber

The application of endophytic extracts on seeds of *L. luteus* significantly improved the fitness of the seedlings that emerged in presence of the pathogen *P. cinnamomi* in terms of root development (Table 2) after 15 days of incubation in the growth chamber.

The seedlings originating from seeds that had been treated with the extract of the endophyte E179 (*E. nigrum*) showed higher roots length in comparison to the controls. However, the number of roots developed was significantly increased by the extract of other endophytes, in this case, *M. hiemalis* (E063), as is displayed in Table 2. The rest of the growth parameters evaluated were not significantly affected by the extract application, regardless of the presence or not of the pathogen.

#### 2.2.2. Effect of the Post-Emergence Application of Extracts on *Lupinus* Seedlings in Greenhouse

None of the endophytic extracts caused apparent plant toxicity as its application in plants non-inoculated with the pathogen did not produce any visible symptom of damage and plant development was completely normal and similar to those of non-treated plants.

The application of endophyte extracts on 30-days-old seedlings of *L. luteus* significantly reduced the disease severity caused by the pathogen *P. cinnamomi* estimated through the area under the disease progress curve (AUDPC). All the endophytic isolates evaluated were able to reduce such an incidence to some extent after one month from the inoculation, although the endophyte E168 (*Fusarium avenaceum*) was the most effective in the pathogen control, reducing by almost 73% the AUDPC (Table 3). Regarding the plant growth traits, the pathogenicity of *P. cinnamomi* was especially evident in the root length and the number of roots, because for both parameters the values obtained from plants inoculated with the pathogen (control plants) were much lower than those of the non-inoculated plants (blank plants). For the root length, the extract of three of the endophytes (E064, E168, and E179) produced not only the reversion of the negative effect caused by the pathogen but the improvement of such parameter, by increasing its value by almost 30% in comparison with the blank plants (Table 3). In the case of the number of roots, the extract of two of the endophytes (E063 and E179) reverted the negative impact caused by *P. cinnamomi*. For these two parameters, the effect of the extracts could be considered protective against the pathogen.

Regarding dry matter (DM) weight, since a significant negative effect of the pathogen was not observed, the protective role of the endophytic extract could not be evaluated. However, the extract of two of the endophytes (E168 and E179) clearly increased the DM weight of the seedling after their application. As regards the most effective extract, that of the endophyte *E. nigrum* (E179), its application increased the shoot DM, root DM, and total DM weight by 63%, 34%, and 53%, respectively, in comparison with the plants without neither pathogen nor extract treatment, and by 80%, 50% and 69%, respectively, in comparison with the non-treated plants inoculated with *P. cinnamomi*. In these cases where *P. cinnamomi* did not exhibit a clear pathogenic behavior, the endophyte extract could be considered to have a plant growth promotion effect, rather than a protective effect.

## 3. Discussion

The filtrates and/or extracts of all the four endophytes evaluated in the present study showed some inhibitory activity against *P. cinnamomi* to a variable degree. The results of in vitro assays to evaluate their effect directly on the pathogen, together with those in planta, allow us to discuss the mechanism involved in such antagonism. In those cases where the filtrate (or extract) inhibited the growth or germination of the pathogen, it may be hypothesized that the antagonistic effect was due to specific metabolites with antimicrobial properties. In particular, this was the case of *D. biseptata* (E064) which completely inhibited the mycelial growth of the pathogen during the whole length of the experiment. Species of the genus *Drechslera* have been found to produce Monocerin, a substance from the chemical family of polyketides, with antifungal activity against *Botrytis cinerea* and *Sclerotinia sclerotiorum* [32]. *D. biseptata* has also been found to produce cellulases and proteases, not antimicrobial substances strictly speaking, but that can be involved in the biocontrol observed against the pathogen as they can help to degrade the cell wall of the pathogen organisms, thus reducing their populations and their infective capacity [33]. The isolation and the identification of the metabolites contained in the filtrates (or extracts) of this endophyte should be further performed to confirm the molecule/s responsible for such an effect.

The endophyte E179 (*E. nigrum*) was also able to limit the in vitro mycelial growth of *P. cinnamomi*. This might presuppose the presence of some substance with antifungal activity in its filtrate (or extract). Several metabolites with antifungal properties have already been identified in *E. nigrum*, such as Epirodin, a polyene with antimicrobial properties against *B. cinerea* [34], and Pretrichodermamide A, a dithiodiketopiperazine derivative that has been found to be antagonistic of *Ustilago maydis* [35]. In both cases, the presence of yellow compounds that diffuse readily into culture media was observed, similar to what we observed in isolates tested in this study. Again, further efforts should be made to identify the substances involved in our case to check if they are the same as those in other studies. However, in the case of the endophyte E179, in addition to the production of substances with antimicrobial properties, its extract also showed a plant growth promotion activity in *L. luteus* seedlings as its application improved growth traits in the plants even in comparison with the plants not inoculated with *P. cinnamomi*. Therefore, different metabolites, other than those with antimicrobial properties, might have been produced by the endophyte to explain such a growth-promoting effect. A growth-promoting effect is a common feature of other biocontrol agents [36]. Other isolates of *E. nigrum* have been found to increase the nutrient uptake of different minerals in plants infected with the endophyte [37] or to increase root growth of sugarcane [38], but in both cases, the living organism was inoculated into plants. In the present study, when the extract has been used instead of the living organism, the improved traits were especially those related to the root growth, which might also explain the higher herbage biomass produced. The presence of the phytohormones-like substance in the extract produced by the endophyte could explain this result. Although the production of this kind of compound by endophytes has already been described [39], further experiments should be performed to identify the actual metabolites involved in this promotion.

The extract of the endophyte E168 (*F. avenaceum*) had also a plant growth promotion effect as its application also improved the growth of the plants non-inoculated with the pathogen. However, in this case, the most prominent result was its biocontrol activity, as it was the extract that produced the highest decrease in the disease severity caused by the pathogen (73%). To explain this biocontrol, the hypothesis of the production of cellulases and proteases could not be applied in this case as the filtrate (or extract) of the endophyte was not able to reduce the mycelial growth of the pathogen. In this case, the application of the filtrate could have activated the defense genes of the plant, a mechanism known as systemic induced resistance, producing antioxidant enzymes, and triggering the metabolism of the methyl-jasmonate, as happens in the case of *Piriformospora indica* when inoculated in rhododendron cultivars towards *P. cinnamomi* and *P. plurivora* [40]. Nevertheless, this hypothesis should be further confirmed.

Some other *Fusarium* species have already shown a certain capacity to control fungal pathogens in plants. Among them, non-pathogenic *F. oxysporum* was found to induce resistance in plants against a wide range of fungal pathogens [41]. Such a capacity to control diseases by *Fusarium* species has been also manifested against oomycetes species, such as *P. capsici* in black pepper [42] or *P. palmivora* in cocoa plants [43]. However, in all these cases the antagonism was produced by using the living organism. Therefore, competition for space and nutrients or hyperparasitism were mechanisms that were indicated to explain the antagonism in addition to the production of antimicrobial substances. Among these antimicrobial substances, *F. oxysporum* has been found to produce several metabolites with bactericidal activity [44]. Those compounds do not seem to be produced by our species/strain or at least they might not have antifungal properties against *Pytophthora* as either the filtrate or the extract from our *Fusarium* endophyte was not able to reduce the growth or germination of the pathogen.

Interestingly, the use of either the filtrate or the extract produced different results in those assays where both fractions were evaluated. That was the case of the MIC assays, where the endophyte E064 (*D. biseptata*), followed by E179 (*Epicoccum nigrum*), showed the greatest antagonistic potential toward *Phytophthora* when the filtrates were used, but the endophyte E063 (*M. hiemalis*) showed the highest inhibitory activity when the extracts were evaluated. However, the other three endophytes also showed significant inhibitory activity. The filtrates are composed of the culture medium (YMB) where the fungi were growing, and the metabolites produced by them. Therefore, in filtrates there can be many compounds that can interfere with fungal metabolites, while in the extracts, many of these other substances are removed during the extraction process. Moreover, in the extracts, the eventual metabolites contained might be more concentrated and purer. Not all the metabolites might be similarly affected by an eventual interference of these other substances. This could explain why this species did not show a positive response when the filtrates were used, maybe because its metabolites were more susceptible to eventual interferences with other substances, but it did when such substances were eliminated. This hypothesis is supported by the fact that in the rest of the assays where the extracts were used, this endophyte (*M. hiemalis*) showed a certain biocontrol capacity against the pathogen. This was the case with the *L. luteus* seed germination, which was improved by the endophyte extract, as well as the number and length of roots in the seedlings treated with this fungus extract. The extract of this endophyte was able also to reduce de disease severity caused by *P. cinnamomi* in *L. luteus* seedlings. This species had already been found to limit the growth in vitro of other *Phytophthora* species, such as *P. infestans* [45], and to show antagonistic properties against other fungal pathogens, such as *Thielaviopsis paradoxa*, in date palm [46]. In those cases, the living organism was used for the experiments, but according to our results, in which only the extracts were used, it could be thought that the responsibility for such an antagonism could be one or several metabolites produced by the endophyte.

## 4. Materials and Methods

### 4.1. Fungal and Plant Material

Four endophytic fungi were selected for this study. These endophytes had been previously isolated from healthy plants collected from *dehesas* of Extremadura, Spain, and identified as *Mucor hiemalis* Wehmer (lab code E063, GenBank accession no. KP899388); *Drechslera biseptata* (Sacc. and Roum.) M.J. Richardson and E.M. Fraser (lab code E064, Gen-Bank accession no. KP698352); *Fusarium avenaceum* (Fr.) Sacc. (lab code E168, GenBank ac-cession no. KP698339); and *Epicoccum nigrum* Link (lab code E179, GenBank accession no. KP698340). Such identification was achieved at a molecular level by the sequencing of their ITS region and the subsequent comparison with sequences from both GenBank and UNITE databases using a BLAST search [47]. A more thorough explanation of the species assignation, among other relevant aspects of fungal identification, can be consulted in Lledó et al. [48] and Santamaría et al. [49]. The selection was made considering several parameters such as the frequency of isolation from the original plant hosts or the observation of some kind of interesting bioactivity in previous assays. In this sense, several studies concerning the effect of these fungi had already been conducted on different pasture hosts, such as *Poa pratensis* and *Trifolium subterraneum* [50], and other legume species such as *Ornithopus compressus* [51]. These and other results related to antimicrobial potential [24] were used for the selection.

The pathogen used, a single morphologically identified *P. cinnamomi strain*, isolated from roots of a *Quercus ilex* tree in Valverde de Mérida, SW Spain (38° 55′ N, 6° 11′ W), had been shown to be highly virulent to seedlings of *Q. ilex* and *Castanea sativa* [52]. The *P. cinnamomi* strain was grown on potato dextrose agar (PDA) and maintained at 23 °C (39 g PDA L^−1^) for mycelial growth. To obtain spores, two plugs (ø = 5 mm) from 7-day cultures growing on PDA were sown on V8-agar plates (900 mL distilled water; 100 mL V8^®^ juice; 1.5 g CACO_3_; 15 g agar) and incubated at 22 °C. After 5 days, the plates were inundated with distilled water and, after 4 h, with 1% non-sterile soil extract for 24 h. Finally, to release zoospores the plates were incubated at 4 °C for 3 h. The zoospore concentration was calculated using a Neubauer chamber. Finally, commercial seeds of *Lupinus luteus* L. (cv. Tremosilla) were used for the in planta experiments.

### 4.2. Culture Conditions and Crude Extract Production from Fungal Endophytes

Each endophyte was incubated in three 500 mL Erlenmeyer flasks containing 250 mL of yeast malt broth, YMB (in [g L^−1^]: yeast extract: 6; malt extract: 10; D-glucose: 6; pH 6.3) at 23 °C and 140 rpm in a thermoshaker (Orbital Shaker Incubator COMECTA 1102, Barcelona, Spain). Two days after the total consumption of the glucose in the medium, the fungal culture was vacuum filtered by using sterile paper discs (ø = 0.2 μm) to separate the mycelium from the liquid filtrate containing the secondary metabolites [53].

Additionally, from part of these filtrates, the crude extract was obtained by following the methodology described by Halecker et al. [54]. Thus, the filtrate was mixed with an equal amount of ethyl acetate, shaken vigorously for 2 min, and poured into a separatory funnel. After the separation of the two phases, a small amount of sodium sulfate was added to the organic phase, to remove possible water residues, and filtered through filters 0.16 mm ø (MN 615 ¼). The sample was then evaporated on a rotary evaporator to eliminate the ethyl acetate (Hei-Vap ML/G1).

### 4.3. Effect of the Fungal Filtrates on the Pathogen Mycelial Growth In Vitro

The inhibitory effect of the fungal filtrates was first determined in vitro against the mycelial growth of *P. cinnamomi*. For this, Petri dishes were prepared by adding 2 mL of the fungal filtrates to 18 mL of sterilized PDA medium, ensuring uniform mixing before the agar solidification. Petri dishes with 2 mL of sterilized distilled water instead of the filtrates were used as control. Once cooled, an actively growing plug of mycelium (ø = 5 mm) of the pathogen was placed in the center of each Petri dish [55]. The samples were kept in the growth chamber at 23 °C for 72 h and every 12 h, four colony length measurements per repetition (one for each of the radii that limited the quadrants) were taken. The experiment was carried out in quadruplicate.

### 4.4. Minimum Inhibitory Concentration In Vitro of the Filtrates and Extracts

The minimum inhibitory concentration (MIC) of the filtrates was determined against *P. cinnamomi* by using a broth microdilution assay in 96-well plates, as described by Halecker et al. [49]. First, 130 µL of a freshly prepared spore solution in YMB (6.7 × 10^5^ spores mL^−1^) of the pathogen was added to each well. Then, 20 µL of the undiluted filtrate was added in the first row, and a 50% serial dilution for the following rows was applied. To check the validity of results, a positive control, with (1.5 mg mL^−1^) cycloheximide, and a negative control, with 20 µL sterilized distilled water, were included. Plates were kept at 30 °C and 600 rpm for 48 h before evaluating the results. The procedure to evaluate the antimicrobial activity of the crude extracts was identical to that followed for the filtrates. In this case, 20 µL (300 µg mL^−1^) per well of each extract was added in the first row and methanol was used as a positive control. All tests were performed in triplicate.

### 4.5. In Planta Experiments

In order to evaluate the efficacy of the crude extracts to protect *L. luteus* (Tremosilla) plants against *P. cinnamomi*, two greenhouse experiments were performed, one applying the fungal extracts in pre-emergence in the seeds, and the other one, applying the extracts in post-emergence in the seedlings. For both experiments, lupine seeds were previously surface sterilized in a solution of 2% sodium hypochlorite, and then washed 3 times with sterile distilled water [56]. The sterilized seeds were sown in 7 × 7 × 6 cm plastic pots, with a mixture of 1:1 (*v*:*v*) substrate and perlite (pH 7.00 ± 0.50; EC 1.50 ± 0.10 dS m^−1^; Organic matter 60.0 ± 2.0%; N 1.29 ± 0.08%; P_2_O_5_ 0.58 ± 0.05%; K_2_O 1.25 ± 0.10%). During the greenhouse tests, plants did not receive any fertilizer. The climatic conditions during the experiments are summarized in Figure 3.

For the pre-emergence test, the sterilized seeds were immersed in the fungal extracts (3 mg mL^−1^) for 6 h in order to assure the metabolites’ absorption. A negative control of sterile distilled water, the solvent used for the extracts, was included. After this period, 30 seeds per treatment were sown in independent pots with the substrate and perlite mixture previously inoculated with a 100 mL L^−1^ soil solution with the pathogen (2 × 10^4^ zoospores of *P. cinnamomi* mL^−1^). A blank treatment not inoculated either with the pathogen was also included in the experiment. After sowing, pots were kept under greenhouse conditions for 15 days, during which the germination was controlled daily. After this period, in five plants randomly harvested per treatment, the aerial and root elongation, the number of roots, and the dry weight of each part (root and herbage) and total were determined. The test was carried out in triplicate.

For the post-emergence test, the sterilized seeds were sown in pots and the plants were allowed to grow freely for a month before any treatment, with the necessary irrigation to keep them at field capacity. After this period, 100 mL L^−1^ of soil of the pathogen solution (2 × 10^4^ spores of *P. cinnamomi* mL^−1^) were added to each pot, and then the endophyte treatment was carried out by spraying a dose of 1 mL [3 mg mL^−1^] of the crude extract, at a rate of 1 mL per pot. A control treatment with sterilized distilled water was included in the experiment, as well as a blank treatment, not inoculated either with the pathogen. Disease severity measurements began one week after the pathogen inoculation and continued weekly for a month. Disease severity was estimated through the percentage of the plant affected by symptoms of disease visually observed in each pot (symptoms considered were yellowing, drying, rotten leaves, and/or blackish spots). According to that percentage, five levels of severity were established: 0—Healthy plant; 1—slight signs of disease (affecting less than 50% of the plant); 2—symptoms of disease (affecting more than 50% of the plant); 3—senescence or obvious signs of decay; 4—death. The area under the disease progress curve (AUDPC) for each pot was built through the assigned severity values, as the sum of the area of the corresponding trapeziums, considering each period between two consecutive measurements as a unit. After the last measurement, the herbage and roots of five plants per treatment were collected for lab analysis. There, the number of roots and the root length were first recorded. Then, samples were oven-dried at 60 °C until constant weight, obtaining the weight of the herbage and root dry matter.

In both tests, blank treatments, plants non-inoculated with the pathogen and without any endophytic extract treatment, were included in the analyses in order to evaluate the virulence of the pathogen. Additionally, although not included in the statistical analyses, plants non-inoculated with the pathogen were treated with each endophytic extract, in order to verify that crude extracts did not have any negative effect on plant development.

### 4.6. Statistical Analysis

The effect of the filtrates on the mycelial growth of *P. cinnamomi* and the effect of the extracts on the seed germination of *L. luteus* inoculated with the pathogen were analyzed by means of mixed-design models, in this case split-plot ANOVAs, including the main-plot factor ‘measurement’ (2017/18 and 2018/19) and the subplot factor ‘treatment’, and their interactions in the model. For the *in planta* experiments, the effect of the crude extracts on herbage and root length, herbage and root dry matter, and number of roots of the harvested plants were evaluated by one-way ANOVA. For these analyses, when significant differences were found in ANOVAs, means were compared using Fisher’s protected least significant difference (LSD) test at *p* ≤ 0.05. Assumptions of normal distribution and homoscedasticity were assured by Shapiro–Wilk and Levene’s tests, respectively.

## 5. Conclusions

The results presented here support the hypothesis that in many cases the protective effect produced by fungal endophytes in their host against fungal pathogens, in this case, *P. cinnamomi*, is due to metabolites. This fact might allow using these compounds, instead of the living organism, for the biocontrol strategies against the pathogen. The possibility to use extracts or metabolites instead of living organisms has been envisaged and explored with other biocontrol agents, such as species of *Trichoderma* [57]. The handling and application of these products are much easier than in a living organism, and the expected bio-protection might not be so dependent on the environmental conditions, supposedly obtaining thus more consistent results. This is the first step for the further development of a new bio-product with commercial significance. Although the extracts could be used directly in the formulation of the product, further isolation and identification of the metabolites specifically involved in the biocontrol may optimize the efficiency of its application. The application range, hosts, and pathogens, in which the product might be effective should be also evaluated.

## Figures and Tables

**Figure 1 plants-11-01455-f001:**
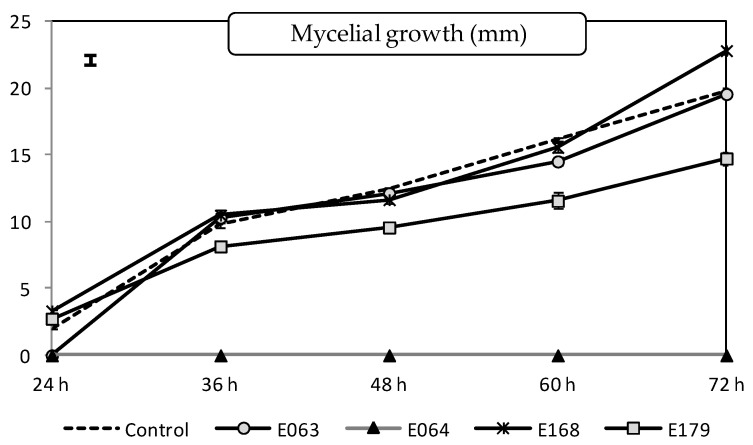
Effect of endophytic filtrates on the mycelial growth in vitro of *Phytophthora cinnamomi* colony over five measurements (after 24, 36, 48, 60, and 72 h). Values are expressed as mean (*n* = 16) ± standard error (error bars). The bar in the upper-left corner indicates the least significant difference (LSD) at α = 0.05.

**Figure 2 plants-11-01455-f002:**
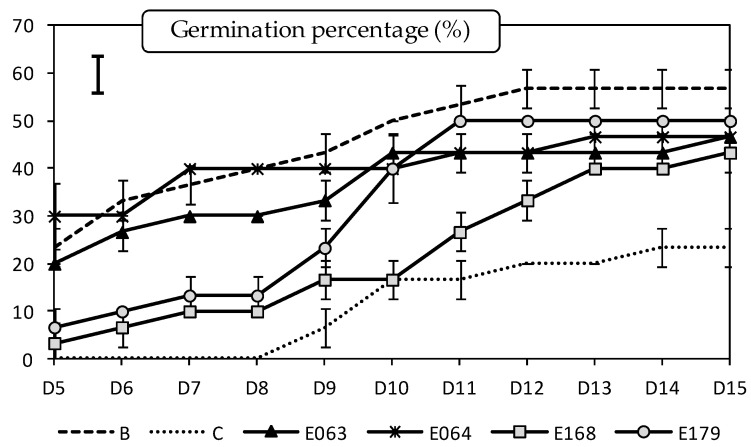
Effect of endophytic extracts on the germination percentage of *Lupinus luteus* seeds inoculated with *Phytophthora cinnamomi* over 11 measurements (from 5 to 15 days). Values are expressed as mean (*n* = 16) ± standard error (error bars). The bar in the upper-left corner indicates the least significant difference (LSD) at α = 0.05. B (blank): seedling receiving no extract nor *Phytophthora* inoculation. C (control): *Phytophthora*—inoculated seeds receiving no extract.

**Figure 3 plants-11-01455-f003:**
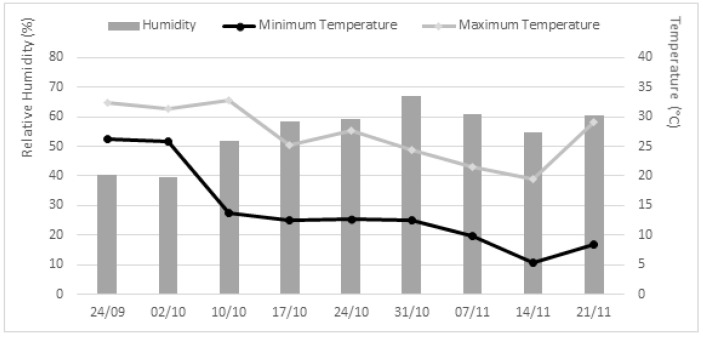
Temperatures and relative humidity values in the greenhouse during the experiments.

**Table 1 plants-11-01455-t001:** Minimum inhibitory concentration (MIC) values of the filtrates (expressed in ratio for one unit) and extracts (expressed in μg mL^−1^) of each endophyte against *Phytophthora cinnamomi* in comparison with a positive control using cycloheximide (C) and a negative one using yeast extract medium (B).

Treatments	MIC	
	Filtrate Dilution	Extract
B	0	0.00
C	-	2.34
E063	0	150.00
E064	1/2	300.00
E168	0	300.00
E179	1	300.00

**Table 2 plants-11-01455-t002:** Effect of the pre-emergence application of endophytic extracts on different growth traits (Sl: shoot length; Rl: root length; number of leaves and roots; DMs: shoot dry matter weight; DMr: root dry matter weight; DMt: total dry matter weight) in *Lupinus* seedlings inoculated with *Phytophthora cinnamomi* in vitro. Values are expressed as mean (*n* = 5) ± standard error. A summary of ANOVAs is also given at the bottom of the table (*df*: degree of freedom; *F*: *F*-value; and *p*-value).

Treatments	Sl (cm) ^1^	Rl (cm)	Leaves #	Roots #	DMs (mg)	DMr (mg)	DMt (mg)
B ^2^	7.7 ± 1.0	5.5 ± 1.0 c	3.5 ± 0.6	10.8 ± 1.0 b	72.0 ± 7.4	24.8 ± 6.4	96.8 ± 13.5
C ^3^	6.3 ± 0.3	6.4 ± 0.1 bc	3.0 ± 0.0	9.0 ± 0.5 bc	53.0 ± 5.2	16.0 ± 3.8	69.0 ± 9.0
E063	7.9 ± 0.7	6.9 ± 0.9 bc	3.8 ± 0.6	13.0 ± 0.7 a	63.5 ± 4.5	18.3 ± 2.5	81.8 ± 4.9
E064	7.5 ± 0.8	7.7 ± 0.3 ab	2.8 ± 0.3	10.0 ± 0.8 b	62.8 ± 10.5	16.5 ± 5.3	79.3 ± 15.1
E168	5.5 ± 0.7	7.0 ± 0.6 abc	3.5 ± 0.6	7.8 ± 0.9 c	56.8 ± 7.9	12.5 ± 0.6	69.3 ± 7.8
E179	7.5 ± 0.5	8.6 ± 0.4 a	3.0 ± 0.3	9.5 ± 0.3 bc	66.0 ± 5.1	19.8 ± 0.3	85.8 ± 4.9
*df*	5	5	5	5	5	5	5
*F*	2.24	3.63	0.97	8.02	1.21	1.53	1.48
*p-value*	0.0943	0.0191	0.4613	0.0004	0.3451	0.2316	0.2448

^1^ For each parameter, different letters indicate significant differences according to LSD (least significant difference) test at α = 0.05. When letters do not appear, no significant effect (*p* > 0.05) was found according to ANOVA. ^2^ B: blank. Seedling receiving no extract nor *Phytophthora* inoculation. ^3^ C: control. *Phytophthora*—inoculated seedling receiving no extract.

**Table 3 plants-11-01455-t003:** Effect of the post-emergence application of extracts of endophytes on the disease severity, expressed in terms of the area under the disease progress curve (AUDPC), and different growth traits (Rl: root length; number of roots; DMs: shoot dry matter weight; DMr: root dry matter weight; DMt: total dry matter weight) in *Lupinus* plants inoculated with *Phytophthora cinnamomi* under greenhouse conditions. Values are expressed as mean (*n* = 5) ± standard error. A summary of ANOVAs is also given at the bottom of the Table (*df*: degree of freedom; *F*: *F*-value; and *p*-value).

Treatments	AUDPC ^1^	Rl (cm)	Roots #	DMs (mg)	DMr (mg)	DMt (mg)
B ^2^	0.0 ± 0.0 d	19.6 ± 2.7 c	14.2 ± 0.4 ab	456.3 ± 51.8 c	271.9 ± 26.0 bc	728.3 ± 69.8 b
C ^3^	168.0 ± 0.0 a	14.7 ± 1.4 d	6.2 ± 1.0 d	415.4 ± 42.6 c	242.9 ± 17.6 c	658.3 ± 44.2 b
E063	103.2 ± 5.4 b	20.2 ± 0.6 bc	14.6 ± 0.6 a	427.0 ± 33.7 c	266.7 ± 21.2 c	693.7 ± 54.1 b
E064	105.6 ± 13.0 b	25.9 ± 0.9 a	12.8 ± 0.4 bc	455.6 ± 34.5 c	259.8 ± 41.1 c	715.4 ± 73.6 b
E168	45.6 ± 10.7 c	25.2 ± 1.9 a	11.6 ± 0.6 c	637.3 ± 37.9 b	353.9 ± 51.0 ab	991.3 ± 81.8 a
E179	81.6 ± 14.3 b	24.2 ± 0.8 ab	13.6 ± 0.4 ab	746.4 ± 28.2 a	365.0 ± 20.0 a	1111.4 ± 30.5 a
*df*	5	5	5	5	5	5
*F*	47.53	9.44	32.29	15.35	3.34	11.56
*p-value*	<0.0001	<0.0001	<0.0001	<0.0001	0.0197	<0.0001

^1^ For each parameter, different letters indicate significant differences according to LSD (least significant difference) test at α = 0.05. When letters do not appear, no significant effect (*p* > 0.05) was found according to ANOVA. ^2^ B: blank. Plants receiving no extract nor *Phytophthora* inoculation. ^3^ C: control. *Phytophthora*—inoculated plants receiving no extract.

## Data Availability

All data are included in the present study.

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
