# Peer review of "Protective Effects of Filtrates and Extracts from Fungal Endophytes on Phytophthora cinnamomi in Lupinus luteus"

_plants, 2022, doi:10.3390/plants11111455_

Round 1

Reviewer 1 Report

The Authors explored the possibility to use culture extracts of fungal endophytes to control the oomycete Phytophthora cinnamomi a serious pathogen of Lupinus luteus. The idea of use metabolites or culture extracts instead of living fungi as biocontrol agents is not completely original but has not been widely explored, especially for fungal endophytes.

This article reports results of preliminary in vitro and in planta tests of the antioomycete activity of raw culture filtrates or culture extracts of fungal endophytes recovered from Lupinus luteus, which is a pulse crop widely used in dehesas, southern Spain. Phytophthora cinnamomi was used as test organism.

The experimental design is relatively simple but adequate, considering the test is preliminary, and results are clearly presented. The Discussion could be improved also introducing pertinent literature concerning other potential biocontrol fungi. I added few references but if the Authors want to include additional relevant references they are welcome.

For additional comments and detailed edits see notes in the text (attached file)

Author Response

Response: The discussion has been improved according to the reviewer comments and the references included. The rest of the aspects indicated by the reviewer in the attached file has also been considered conveniently.

Comment on page 7, lines 262-264. IN MY OPINION THIS EXPLANATION IS NOT CONVINCING AS THE GROWTH PROMOTING EFFECT SHOULD BE INFERRED BY COMPARING THE TREATED PLANTS WITH THE NON INOCULATED CONTROL PLANTS (POSITIVE CONTROL). PLEASE COMMENT ON THIS.

Response: We agree with the reviewer as it is true that the growth promotion effect is inferred by the comparison with not inoculated seedlings. This part has been revised and re-written accordingly.

Reviewer 2 Report

How did you identify the pathogen Phytophthora cinnamomi? Is this based on a morphological study? Did you submit the sequence of these isolates to GenBank?

Also, generally, the species such as Fusarium avenaceum are pathogens to many plants; how did you confirm these isolates are non-pathogenic?

I also can't entirely agree with using the ITS region alone; you could be identified the Fusarium isolate to the species level. this needs more gene regions

Author Response

How did you identify the pathogen Phytophthora cinnamomi? Is this based on a morphological study? Did you submit the sequence of these isolates to GenBank?

Response: A single morphologically identified P. cinnamomi strain, isolated from roots of a Quercus ilex tree in Valverde de Mérida, SW Spain (38° 55′ N, 6° 11′ W), and highly virulent to seedlings of Q. ilex and C. sativa was used. All the information regarding this strain can be obtained from Camisón et al. (2019). This information has been included in the manuscript.

Also, generally, the species such as Fusarium avenaceum are pathogens to many plants; how did you confirm these isolates are non-pathogenic?

Response: The Fusarium avenaceum isolate was not directly inoculated in plants, but the extract it produced was used in the treatments. Any toxicity of this extract (and from the rest of endophytes) was evaluated by its application in plants non-inoculated with the pathogen, in order to verify that crude extracts did not have any negative effect on plant development. None of the endophytic extracts caused apparent plant toxicity as its application in plants non-inoculated with the pathogen did not produce any visible symptom of damage and plant development was completely normal and similar to those non-treated plants. This fact has been described more clearly in the revised version of the manuscript, in the M&M and Results sections.

I also can't entirely agree with using the ITS region alone; you could be identified the Fusarium isolate to the species level. this needs more gene regions

Response: For species level analyses, most studies dealing with fungal endophytic assemblages have used information from the internal transcribed spacer region (ITS) (Sánchez Márquez et al., 2007; Botella & Diez, 2011; Ek-Ramos et al., 2013; Zhang et al., 2014), which has been shown to be a very useful locus for this purpose. In clear agreement with that, Schoch et al. (2012) proposed the ITS region as a universal DNA barcode marker for fungi, as they found the best results when using this region in comparison with other commonly used ones, such as LSU (ribosomal large subunit rRNA gene), SSU (ribosomal small subunit rRNA gene), and RPB1 (the largest subunit of RNA polymerase II). It is true, however, that several other studies have evidenced the relatively low resolution of ITS sequencing, and the difficulty in aligning ITS sequences, which restricts its utility for phylogenetic reconstruction, especially in mushroom fungi (Frøslev et al., 2005). Errors in the identification of several fungal groups have been also reported when using ITS region, especially in species-rich genera with shorter amplicons, such as Cladosporium, Penicillium, and Fusarium (O’Donnell & Cigelnik, 1997; Schubert et al., 2007). In our study, in the case of Fusarium, in addition to the ITS analyses, the morphology of the spores was also analyzed and the species identification obtained by following identification keys completely matched with the identification obtained by ITS sequencing. This explanation can be found in Lledó et al. (2016) and Santamaría et al. (2018), such as it is indicated in the M&M section of the manuscript.

Reviewer 3 Report

I recommend the publication the article because scientific experimentation concerning biocontrol is topical and of particular interest.
The aim and objectives of the article have been stated and are very interesting. The use of metabolites obtained from endophytic fungi for plant protection is an important topic, especially for reducing synthetic products in agriculture. The work carried out is undoubted of international interest, and the format applied is undoubtedly suitable for a research article. The result is original, of particular interest, and can stimulate research on this topic. The article's length is good for the journal, and the graphs and tables are clear and easy to understand. The conclusion summarises the objectives of the work and its prospects.

Author Response

Response: Thank you very much for the so supportive comments.